# A Direct Interaction between Cyclodextrins and TASK Channels Decreases the Leak Current in Cerebellar Granule Neurons

**DOI:** 10.3390/biology11081097

**Published:** 2022-07-23

**Authors:** Rafael Zúñiga, Daniel Mancilla, Tamara Rojas, Fernando Vergara, Wendy González, Marcelo A. Catalán, Leandro Zúñiga

**Affiliations:** 1Laboratorio de Fisiología Molecular, Escuela de Medicina, Universidad de Talca, Talca 3460000, Chile; rafaelzunigah@gmail.com (R.Z.); daniel.mancilla@utalca.cl (D.M.); trojas16@alumnos.utalca.cl (T.R.); fer.vergara2@gmail.com (F.V.); 2Centro de Nanomedicina, Diagnóstico y Desarrollo de Fármacos (ND3), Escuela de Medicina, Universidad de Talca, Talca 3460000, Chile; 3Instituto de Investigación Interdisciplinaria, Vicerrectoría Académica, Universidad de Talca, Talca 3460000, Chile; 4Center for Bioinformatics and Molecular Simulations (CBSM), Universidad de Talca, Talca 3460000, Chile; wgonzalez@utalca.cl; 5Facultad de Medicina, Instituto de Fisiología, Universidad Austral de Chile, Valdivia 5090010, Chile; marcelo.catalan@uach.cl

**Keywords:** cyclodextrin, K2P channels, K^+^ leak currents, cerebellar granule neurons

## Abstract

**Simple Summary:**

Cyclodextrins are cyclic oligosaccharides used to deplete cholesterol from cellular membranes. The effects of methyl-β-cyclodextrin (MβCD) on cellular functions originate principally from reductions in cholesterol levels. In this study, using immunocytochemistry, heterologous expression of K2P channels, and cholesterol-depleting maneuvers, we provide evidence of expression in cultured rat cerebellar granule neurons (CGNs) of TWIK-1 (K2P1), TASK-1 (K2P3), TASK-3 (K2P9), and TRESK (K2P18) channels and their association with lipid rafts using the specific lipids raft markers. In addition, we show a direct blocking with MβCD of TASK-1 and TASK-3 channels as well as for the covalently concatenated heterodimer TASK-1/TASK-3.

**Abstract:**

Two pore domain potassium channels (K2P) are strongly expressed in the nervous system (CNS), where they play a central role in excitability. These channels give rise to background K^+^ currents, also known as IK_SO_ (standing-outward potassium current). We detected the expression in primary cultured cerebellar granule neurons (CGNs) of TWIK-1 (K2P1), TASK-1 (K2P3), TASK-3 (K2P9), and TRESK (K2P18) channels by immunocytochemistry and their association with lipid rafts using the specific lipids raft markers flotillin-2 and caveolin-1. At the functional level, methyl-β-cyclodextrin (MβCD, 5 mM) reduced IK_SO_ currents by ~40% in CGN cells. To dissect out this effect, we heterologously expressed the human TWIK-1, TASK-1, TASK-3, and TRESK channels in HEK-293 cells. MβCD directly blocked TASK-1 and TASK-3 channels and the covalently concatenated heterodimer TASK-1/TASK-3 currents. Conversely, MβCD did not affect TWIK-1- and TRESK-mediated K+ currents. On the other hand, the cholesterol-depleting agent filipin III did not affect TASK-1/TASK-3 channels. Together, the results suggest that neuronal background K^+^ channels are associated to lipid raft environments whilst the functional activity is independent of the cholesterol membrane organization.

## 1. Introduction

In excitable cells, K2P channels give rise to background currents, which are open constitutively and voltage independent. In mammals, K2P family is constituted by 15 different members, which also have been found in yeast, plants, zebrafish, nematode, and fruit fly [1,2]. Based on their structural and functional properties, K2P channels are divided in six subfamilies, denoted as TREK (TWIK-related K^+^ channel), TALK (TWIK-related alkaline pH-activated K^+^ channel), TASK (TWIK-related acid-sensitive K^+^ channel), TWIK (tandem of pore domains in weak inward rectifier K^+^ channel), THIK (tandem-pore domain halothane-inhibited K^+^ channel), and TRESK (TWIK-related spinal cord K^+^ channel) [1,3,4,5].

A K2P channel subunit is conformed by four transmembrane domains and two pore forming domains, in tandem [6,7,8,9]. A functional channel is composed of two identical (homodimer) or different subunits (heterodimer) [10,11,12,13,14,15,16,17]. Thus, the heterodimeric configuration of K2P channels increases the functional diversity, versatility, and dynamic adaptation [5,13].

K2P channels are highly regulated by different stimuli including kinases, phospholipids, G proteins, internal and external pH, mechanical force, protein–protein interactions, and volatile anaesthetics [1,4,18,19,20,21]. Additionally, the extracellular pH modulates K2P channels opening by acting on the upper gate [20,22,23,24].

In CGN, K2P channels are highly expressed and generate IK_SO_ (for standing-outward K^+^ current) [13,25,26,27]. The IK_SO_ activity is modulated by the extracellular pH changes [27,28,29], where the acidosis decreases the IK_SO_ current and has been associated with an increase in excitability [30,31]. K2P channels are identified as critical players in a variety of clinically relevant processes including neuroprotection, general anaesthesia, pain, depression, and cancer [18,32,33,34,35,36], thus representing an important potential therapeutic target for future clinical research.

K2P channels control the resting membrane potential of neurons, regulating the neuronal excitability. Cerebellar granule neurons play a central transduction role in cerebellar function [13,27,37]. The relevance of the IK_SO_ current in the regulation of CGN excitability has been widely demonstrated [13,26,27] and suggests that the increase in spontaneous excitatory postsynaptic currents (EPSCs), displayed by cerebellar Purkinje cells by an acetylcholine synaptic release, occurs as a result of IK_SO_ current inhibition in CGNs [38]. While GABAergic neurons (Purkinje cells) provide the major output of the cerebellum, glutamatergic interneurons (granule cells) generate an excitatory input in the molecular layer of the cerebellum. Granule cells receive sensory input from mossy fibers and convey the information to Purkinje cells via parallel fibers. Furthermore, granule cells exhibit a low frequency of spontaneous firing under in vivo conditions, but are very sensitive to sensory stimulation [39,40].

CGNs display mRNA levels of different K2P channels [41,42]. Of these, in cerebellar granule neurons, the protein expression levels of TWIK-1, TASK-1, TASK-3, and TRESK channels represent the majority component of IK_SO_ current [27]. Moreover, in these cells (CGN), the regulatory mechanism of TWIK-1, TASK-1, and TASK-3 heterodimers has been studied [13]. Furthermore, the pH dependence displayed by the IK_SO_ is consistent with TWIK-1, TASK-1, and TASK-3 channels’ expression [27,43,44,45].

The tight regulation of K2P channels is crucial for the fine tuning of electrical properties of cells. On the other hand, there is growing evidence supporting that membrane rafts are key ion channel regulators [46,47,48,49]. Thus, the lipidic environment might modulate K2P ion channels by a structural and/or functional modulation through direct protein–lipid interactions or by influencing the biophysical properties of the plasma membrane [46].

Here, we evaluated the expression of K2P channels and co-localization of K2P channels with lipids’ raft protein markers in cultured CGNs. Additionally, we investigated the functional effect of cholesterol depletion on the channel activity. We found that methyl-β-cyclodextrin (MβCD) decreased the background current activity by a direct interaction with TASK channels. Nevertheless, a direct effect due to the cholesterol-depleting treatment on the K2P channel function was ruled out. Our results indicate that neuronal K2P channels (TWIK-1, TASK-1, TASK-3, and TRESK) interact with the lipid rafts, but their functional activity is independent of the membrane composition and organization of cholesterol domains.

## 2. Materials and Methods

### 2.1. Cerebellar Granule Neuron Cultures

Cerebellar granule neurons (CGNs) were dissociated from 7- to 9-day-old Sprague–Dawley rat cerebellum and purified as previously described [50,51]. Briefly, the cerebella were triturated to dissociate the neurons and plated onto coverslips treated with poly-L-lysine (1 µg/mL) (Invitrogen Life Technologies, Carlsbad, CA, USA) at a density of 2.5 × 105 cells/cm^2^. Cells were then incubated at 37 °C in a 5% CO_2_ in DMEM medium (Thermo Fisher Scientific, Waltham, MA, USA) supplemented with 10% FCS (fetal calf serum), 25 mM KCl, 39 mM glucose, 5 mM glutamine, and 1% penicillin and streptomycin. After 4 days, the culture medium was exchanged. The experiments with CGNs were performed between 7 and 14 days in culture. The experimental procedures were approved by the Institutional Bioethics Committee (CIECUAL) of the University of Talca (Code: CIECUAL-UTALCA 22-01).

### 2.2. Immunocytochemistry

The K2P channels’ co-localization in cultured Cerebellar granule neuron was studied by immunofluorescence. Briefly, cultured Cerebellar granule cells were fixed with 1× phosphate-buffered saline (PBS) supplemented with 4% paraformaldehyde (PFA) for 20 min at room temperature and were blocked and permeabilized with 2% BSA in PBS containing 0.1% Triton X-100 for 30 min. Fixed CGN neurons were double-labeled by incubation with goat polyclonal antibodies against TWIK-1, TASK-1, TASK-3, or TRESK (1:100 for sc-11481, sc-32067, sc-11317, and sc-51240; Santa Cruz Biotechnology, Dallas, TX, USA) and rabbit polyclonal antibodies against flotillin-2 (1:100, sc-25507; Santa Cruz) or caveolin-1 (1:100, ab18199; Abcam, Cambridge, MA, USA) overnight at 4 °C. Primary antibody incubation was followed by incubation with an Alexa Fluor^®^ 488 or Alexa Fluor^®^ 594 secondary antibody (1:1000 for ab150073 and ab150132; Abcam) for 1 h at room temperature. Controls were carried out with CGN and prepared under identical conditions, with the omission of primary antibody, and no fluorescence signal was detected (data not shown). Co-localization of fluorescent labelling was visualized with a laser scanning confocal microscope (Zeiss LSM-700 microscope, Oberkochen, Germany) using a 40× oil-immersion lens and Photometrics SenSys camera (Photometrics, Tucson, AZ, USA). The images were analyzed with ImageJ (National Institutes of Health, Bethesda, Rockville MD, USA; http://rsb.info.nih.gov/ij/ (accessed on 20 July 2022)). Quantitative co-localization analysis of fluorescence microscopy images was carried out using the co-localization threshold plugin [52] of ImageJ, which uses the threshold algorithm of Costes et al. [53]. All experiments were performed less three times and with independent cell cultures.

### 2.3. Electrophysiological Recordings in CGN Neurons

Macroscopic currents obtained from CGNs were studied using the whole-cell patch-clamp configuration with a PC-501A amplifier (Warner Instruments, Hamden, CT, USA), as described previously [27]. The pCLAMP10 with an acquisition card (DigiData 1440, Molecular Devices, San Jose, CA, USA) was used for voltage protocols and data acquisition, Glass microelectrodes (3–5 MΩ) were made from borosilicate capillaries using P97 Flaming/Brown Micropipette Puller (Sutter Instruments, Novato, CA, USA). The intracellular solution was composed of: 140 mM KCl, 10 mM HEPES, 5 mM EGTA, 2 mM K_2_-ATP, 1 mM MgCl_2_, and 0.5 mM CaCl_2_ and adjusted to pH 7.4 with KOH. The extracellular solution contained (in mM): 120 mM NaCl, 10 mM glucose, 10 mM HEPES, 4 mM KCl, 2 mM MgCl_2_, and 0.5 mM CaCl_2_ and pH 7.4 adjusted with NaOH. To isolate K2P-mediated currents from endogenous sodium currents, the extracellular solution was supplemented with tetrodotoxin (TTX) at 0.2 µM.

### 2.4. HEK-293 Cell Studies

The HEK-293 cell line was obtained from the American Type Culture Collection (Manassas, VA, USA). HEK-293 cells were cultured in DMEM-F12 media (Invitrogen) supplemented with 10% FBS (Thermo Fisher) and 1% penicillin and streptomycin. Cells were grown in a humidified incubator at 37 °C and 5% CO_2_.

For the electrophysiological experiments, HEK-293 cells were transfected with cDNAs encoding human TWIK-1 (Mutant K274Q, an active unsumoylated channel) (NM_002245), TASK-1 (NM_002246), TASK-3 (AF212829), TRESK (NM_181840), and the concatenated construct TASK-1/TASK-3 (a covalently linked heterodimer channel [13]). Co-transfections of plasmids containing cDNAs of interest and a reporter vector encoding the cDNA for green fluorescent protein (GFP) (1–2 μg of DNA plasmid) were achieved with a 3:1 ratio (K2P channel plasmid: GFP plasmid) using Xfect polymer (Clontech, Mountain View, CA, USA). The cells were incubated for 3 h in transfection medium OptiMEM (Invitrogen). After incubation, the medium was exchanged with fresh culture medium and maintained at 37 °C with 5% CO_2_ for 12 h before electrophysiological measurements were made. All reported studies were performed in at least three independent experiments, with replicate transfections in each experiment.

TWIK-1 (Mutant K274Q), TASK-1, TASK-3, and TASK-1/TASK-3 constructs were a kind gift from Dr. Steve Goldstein (University of California, Irvine, CA, USA). The TRESK construct was a generous gift from Dr. Péter Enyedi (Semmelweis University, Budapest, Hungary).

### 2.5. Electrophysiology

For whole-cell recordings, HEK-293 cells were transfected with the different K2P wild type or chimeric channels, using a PC-501A patch clamp amplifier (Warner Instruments) and borosilicate glass pipettes as previously described by Zúñiga et al. [23]. The cells were superfused with a solution containing 135 mM NaCl, 10 mM HEPES, 10 mM Sucrose, 5 mM KCl, 1 mM MgCl_2_, and 1 mM CaCl_2_ and adjusted to pH 7.4 with NaOH. The pipette was filled with a solution of 145 mM KCl, 10 mM HEPES, 5 mM EGTA, and 2 MgCl_2_, pH 7.4 adjusted with KOH. Methyl-β-cyclodextrin (MβCD), α-cyclodextrin (αCD) (Sigma-Aldrich, St. Louis, MO, USA), and filipin III (Cayman chemical, Ann Arbor, MI, USA) were dissolved in water or DMSO to obtain 100 mM and 500 µg/mL stock solutions. Working concentrations of 5 mM MβCD, 5 mM αCD, and 5 μg/mL filipin III were then prepared by diluting stock solutions with the bath solutions obtaining the desired concentrations. Cells were held at −80 mV, then currents were recorded using a protocol of 500 ms of duration from −100 to +100 mV with increments of 10 mV. Patch-clamp acquisition and analysis was conducted with pClamp 10 Software (Molecular Devices). Data analysis was performed using SigmaPlot version 12.0 (Systat Software Inc., San Jose, CA, USA).

### 2.6. Molecular Docking

The TASK-1 crystal structure (Protein Data Bank, PDB: 6RV2) was used for docking calculations. The structure of MβCD was obtained from the crystal structure of gastric inhibitory polypeptide receptor (PDBID 2QKH) that was cocrystallized with MβCD [54]. The MβCD and αCD ligands were designed using LigPrep (Schrödinger, LLC, New York, NY, USA, 2017) force field OPLS-2005, with a maintained charge during the parametrization. Then, the compounds were minimized using Macromodel (Schrodinger, LLC, New York, NY, USA, 2017). We performed molecular dockings using Glide software and the standard precision (SP) scoring function to find the better pose of the CD interacting with TASK-1 structure [55]. For the pose’s generation, ten poses were considered per conformer and the strain correction for the GlideScore. Poses were compared and analyzed. The binding sites for MβCD and αCD that shared the most residues in common between TASK-1 and TASK-3 channels were selected.

### 2.7. Statistical Analysis

Data were analyzed with the SPSS software package (SPSS Inc., Chicago, IL, USA). Statistical comparison between groups of data were made using paired Student’s *t*-test. *p* < 0.05 Values were considered as significant. All data shown are mean ± SEM.

## 3. Results

### 3.1. K2P Channels Are Associated with Lipid Rafts

CGNs were used as a model to determine whether K2P channels (TWIK-1, TASK-1, TASK-3, and TRESK) were localized in lipid rafts and the potential regulatory effects due to this localization. To this end, we used immunofluorescence and confocal microscopy to evaluate whether K2P channels co-localize with membrane lipid raft markers flotillin-2 (Figure 1) and caveolin-1 (Figure 2). The co-localization of K2P channels with flotillin-2 and caveolin-1 proteins in CGN can be inferred by merging the green and red channels, in the same focal plane, as shown by the yellow color (Figure 1C,F,I,L and Figure 2C,F,I,L). The correlation between pixel intensity histogram of membrane lipids markers (red channel) and K2P channels (green channel) was analyzed by Pearson’s correlation (coefficient values “1” and “0” correspond to perfect co-localization and completely random uncorrelated distribution, respectively) (Figure 1M and Figure 2M). The Pearson’s coefficient (*R*^2^) for flotillin-2 and K2P channels TWIK-1, TASK-1, TASK-3, and TRESK is 0.611, 0.850, 0.685, and 0.664, respectively (Figure 1M), suggesting significant co-localization. Caveolin-1, another specific marker of lipid rafts, showed a significant co-localization with TWIK-1, TASK-1, TASK-3, and TRESK and Pearson’s coefficient of 0.758, 0.846, 0.619, and 0.771, respectively (Figure 2M).

Double immunostaining analyses revealed a similar extensive distribution and co-localization of TASK-1 channels with the flotillin-2 marker in CGN cells (Figure 1D), as the dominant channels localized in these domains. Additionally, in a descending order, we found a co-localization of TASK-1 > TASK-3 > TRESK > TWIK-1 (Figure 1). Similarly, we evaluated the co-localization of K2P channels with caveolin-1 in CGN cells (Figure 2). Coefficient of determination values (*R*^2^) were TASK-1 > TRESK > TWIK-1 > TASK-3 (Figure 2). Together, the data suggest that at least a fraction of TWIK-1, TASK-1, TASK-3, and TRESK channels are associated with the lipid rafts domain, containing flotillin-2 and caveolin-1 markers, consistent with their presence in lipid raft domains.

### 3.2. Effect of MβCD on Leak Potassium Currents

To determine whether the presence of K2P channels in lipid rafts plays a regulatory role on the channel function, we disrupted lipid rafts by depleting cholesterol with 5 mM methyl-β-cyclodextrin (MβCD). MβCD treatment reduced the IK_SO_ currents by ~40% at –20 mV in CGNs (Figure 3A,B). Consistent with this finding, MβCD treatment increased cell input resistance (R_IN_, *p* < 0.05) (Table 1). Conversely, Table 1 shows that the addition of MβCD did not alter the magnitude of the resting membrane potential.

We hypothesized that cholesterol depletion might affect the K2P channels channel gating or activation. Alternatively, the inhibitory effect mediated by MβCD could be explained by a direct interaction between K2P channels and the cyclodextrin. Therefore, we used a heterologous expression system to evaluate the effect mediated by MβCD on each K2P channel (TWIK-1, TASK-1, TASK-3, and TRESK).

### 3.3. K2P Channels Sensitivity to MβCD

To dissect the contribution of the K2P channels in the reduction of leak potassium currents in response to MβCD, we independently evaluated the effect of MβCD on hTWIK-1, hTASK-1, hTASK-3, and hTRESK activities in HEK-293 cells transiently transfected with plasmids encoding for the above channels. As shown in Figure 4A–I, MβCD significantly reduced the currents mediated by TASK-1 (Figure 4E) and TASK-3 channels (Figure 4G). Conversely, MβCD treatment did not affect TWIK-1 and TRESK-mediated potassium currents (Figure 4C,I, respectively). K^+^ current reduction observed in TASK-1 and TASK-3 channels could be consistent with an effect of MβCD on the cholesterol distribution or, alternatively, with a direct inhibition of TASK channels by this treatment.

### 3.4. TASK-1/TASK-3 Heterodimer Sensitivity to Cyclodextrins and Filipin III

In order to obtain insights into the inhibitory effect mediated by MβCD on TASK-1- and TASK-3-mediated K^+^ currents, we assessed the effect of MβCD on heteromeric TASK-1/TASK-3 channels. This heteromeric configuration, TASK-1/TASK-3, has been studied in native models [13,56] and is a relevant component of IK_SO_ in CGN [13]. As seen in Figure 5B,C, both MβCD and αCD (an inactive cyclodextrin, which does not deplete cholesterol) reduced the potassium currents mediated by TASK-1/TASK-3 concatamers. This effect, mediated by MβCD, is independent of the voltage (Appendix A), as the same degree of inhibition was observed in the whole range of voltages studied. To evaluate if the MβCD-mediated inhibition of TASK currents depended on the cholesterol levels, we assessed changes in the activity of TASK-1/TASK-3 concatamers in response to filipin III (5 μg/mL), another cholesterol depleting agent that is not chemically related to MβCD. Filipin III did not have a significant effect on TASK-1/TASK-3-mediated currents, as shown in Figure 5D, suggesting that cholesterol depletion does not affect the K2P channel. Moreover, the application of 3 mM cholesterol did not produce any changes in TASK-1/TASK-3 currents that were significant (*n* = 4; Appendix A).

### 3.5. Analysis of Cyclodextrins Binding Sites in TASK-1 Channels

To identify the binding site(s) of the cyclodextrins (MβCD and αCD) in TASK-1 channels, molecular docking was performed in the TASK-1 crystal structure (PDB: 6RV2), which identified eight potential residues located in the extracellular cavity very close to the entry to the TASK-1 channel that might be involved in MβCD (Figure 6) and αCD (Figure 7) binding: Glu37, Arg68, Lys70, Gly97, Trp184, Gly203, Asp204, and Lys210 (Figure 6D,E and Figure 7D,E).

To examine other potential direct interactions between CDs (MβCD and αCD) and TASK-1, we performed molecular docking analysis between CDs in the intracellular cavity of TASK-1 channel (Figure 8A–C). Four binding potential residues located in the intracellular cavity were found in close proximity to CDs. They are Glu130, Arg245, Glu252, and Lys255 of the TASK-1 channel (Figure 8D,E). The analyses identified a direct binding involving hydrogen bonds and hydrophobic interactions between CDs and TASK-1 residues.

## 4. Discussion

In rat cerebellar granule neurons, TWIK-1, TASK-1, TASK-3, and TRESK subunits forming homodimers and/or heterodimers account for most of the IK_SO_ [13,27]. The IK_SO_ current is critical for modulation of the neuronal excitability and is regulated by several stimuli as muscarinic inhibition, anaesthetics, pH, and sumo/semp activity [13,25,57].

Its presence in lipid rafts and cholesterol regulates the activity of several ion channels and plasma membrane proteins, in different ways [46,47,48,49]. Here, we examined the localization of K2P channels in lipid rafts and found that TWIK-1, TASK-1, TASK-3, and TRESK channels co-localized with the lipid raft markers flotillin-2 and caveolin-1. The degree of co-localization varied among the channels, but it is clear that at least part of the K2P channels is localized in lipid rafts. The results of expression in CGNs are in line with previous reports showing that K2P subunits were associated with IK_SO_ currents in CGN cells [13,27]. The expression of TWIK-1, TASK-1, TASK-3, and TRESK in lipid rafts suggests that they may share a common structural core of lipid raft association; however, more work will be required to explore this hypothesis further.

MβCD treatment partially decreased the IK_SO_ current in CGN (~40% Figure 3A,B). This effect was accompanied by an increase in the R_IN_, which is also consistent with an increased neuronal excitability. The decrease in the amount IK_SO_ current may be related to an effect of disrupting the lipid rafts by depleting cholesterol with 5 mM MβCD [58]. Thus, cholesterol depletion might affect the K2P channel’s gating. A similar inhibitory effect exerted by MβCD has been shown for several ion channels [59,60,61,62,63,64]. In this regard, a previous study has reported a direct inhibitory effect of CDs on K_V_1.3 channel, which is independent of membrane cholesterol depletion and concomitant alterations in membrane biophysical parameters caused by CDs [65].

Here, we showed that MβCD decreased TASK-1 or TASK-3 channel activity by ~40%. Moreover, no effect of MβCD on TWIK-1 and TRESK channels was observed.

αCD, a cyclodextrin that does not deplete cholesterol from the plasma membrane, showed a similar effect on the tandem dimer channel TASK-1/TASK-3, suggesting a direct effect of cyclodextrins on TASK channels. Another line of evidence that supports the direct effect of cyclodextrin was the treatment with filipin III, which did not show a reduction in TASK-1/TASK-3 mediated currents (Figure 5D). The lack of effect mediated by cholesterol depletion with filipin III on TASK-1/TASK-3 channels suggests that the amount of cholesterol in the plasma membrane does not play a major role in the regulation of TASK-1 and TASK-3 channels. Moreover, cholesterol enrichment of the plasma membrane did not affect TASK-1/TASK-3 channels (Appendix A).

Docking analysis suggested cyclodextrins’ binding sites in the extracellular and intracellular cavities of the TASK-1 channel. We propose that the residues involved in the binding site of CD are Glu37, Arg68, Lys70, Gly97, Trp184, Gly203, Asp204, and Lys210 in the extracellular cavity, and Glu130, Arg245, Glu252, and Lys255 in the intracellular cavity.

Extracellular and intracellular binding sites of cyclodextrins in TASK-3 channels are conserved where residues Glu37, Gly97, Trp184, Gly203, Asp204, Glu130, Arg245, and Glu252 might play a key role in cyclodextrin binding. However, it is clear that a mutagenesis approach followed by electrophysiological recordings evaluating the effect of cyclodextrins on mutated TASK-1 and/or TASK-3 channels is certainly needed to discern the role of the residues in the binding sites of CD.

The finding that MβCD blocks the TASK channels in CGN and the heterologous expression system provides evidence and corroborates the role of TASK-1 and TASK-3 channels’ activity in the reduction of IK_SO_. Depolarization that is accompanied by increased R_IN_ is also consistent with increased neuronal excitability.

We suggest that treatment with MβCD on cerebellar granule neurons increases the excitability by a reduction in IK_SO_, and this effect occurs via a direct interaction with the TASK-1 and TASK-3 channels by a voltage-independent mechanism.

## 5. Conclusions

Our study corroborates the localization in lipid rafts of K2P channels and found that TWIK-1, TASK-1, TASK-3, and TRESK channels co-localized with lipid raft markers. The expression of TWIK-1, TASK-1, TASK-3, and TRESK in lipid rafts suggests that they may share a common structural core of lipid raft association. In addition, we show that MβCD treatment decreased the IK_SO_ current in CGN. This effect was accompanied by an increase in the R_IN_, which is also consistent with an increased neuronal excitability. We suggest that the effect of MβCD treatment on leak potassium currents in CGN cells is by a direct interaction with TASK-1 and TASK-3 channels.

## Figures and Tables

**Figure 1 biology-11-01097-f001:**
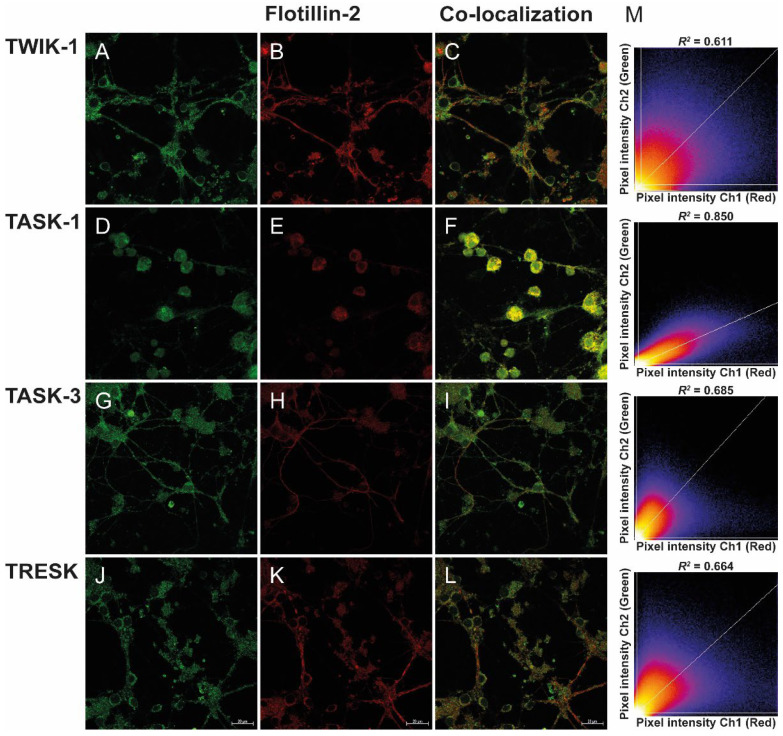
Localization of K2P channels and flotillin-2 in rat cerebellar granule neurons. Immunocytochemical localization of (**A**,**C**) TWIK-1, (**D**,**F**) TASK-1, (**G**,**I**) TASK-3, and (**J**,**L**) TRESK proteins (green fluorescence). (**B**,**E**,**H,K**) Immunofluorescence of flotillin-2 (red fluorescence). (**C**,**F**,**I,L**) K2P channels and flotillin-2 were co-localized on CGN cells (yellow) when the images were merged. The scale bar represents 20 μm. (**M**) Images and Pearson’s correlation analysis of co-localization between K2P channels and flotillin-2. The scatterplot shows a 2D intensity histogram of the red and green pixels. The intensity of pixels above the thresholds (white lines) are co-localized. *R*^2^, Pearson’s correlation index.

**Figure 2 biology-11-01097-f002:**
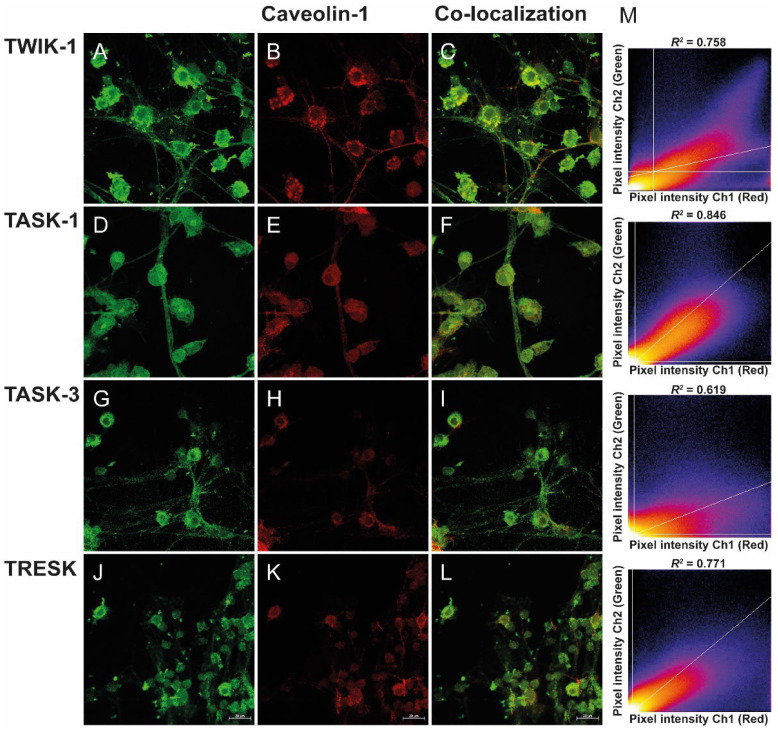
Localization of K2P channels and caveolin-1 in CGNs. (**A**,**C**) Immunocytochemical localization of TWIK-1, (**D**,**F**) TASK-1, (**G**,**I**) TASK-3, and (**J**,**L**) TRESK proteins (green fluorescence). (**B**,**E**,**H**,**K**) Immunofluorescence of caveolin-1 (red fluorescence). (**C**,**F**,**I**,**L**) K2P channels and caveolin-1 were co-localized on CGN cells (yellow) when the images were merged. The scale bar represents 20 μm. (**M**) Representative images and Pearson’s correlation analysis of co-localization between K2P channels and caveolin-1.

**Figure 3 biology-11-01097-f003:**
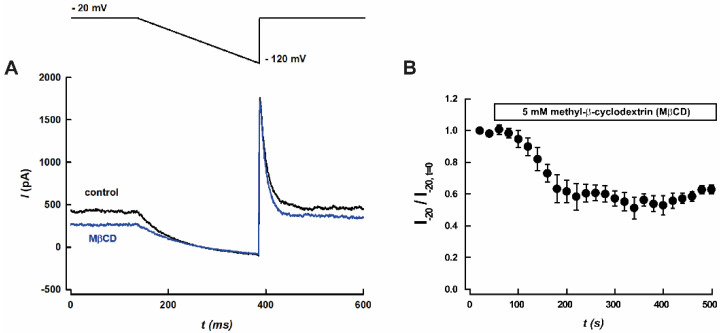
Effect of methyl-β-cyclodextrin (MβCD) on leak potassium currents (IK_SO_) in rat cerebellar granule neurons. (**A**) Representative current traces recorded using the voltage protocol shown in the upper panel, before (black trace) and after application of 5 mM MβCD (blue trace). (**B**) Time course of leak potassium currents (IK_SO_) measured at −20 mV, before and after application of 5 mM MβCD. Results are shown as means ± SEM (*n* = 5).

**Figure 4 biology-11-01097-f004:**
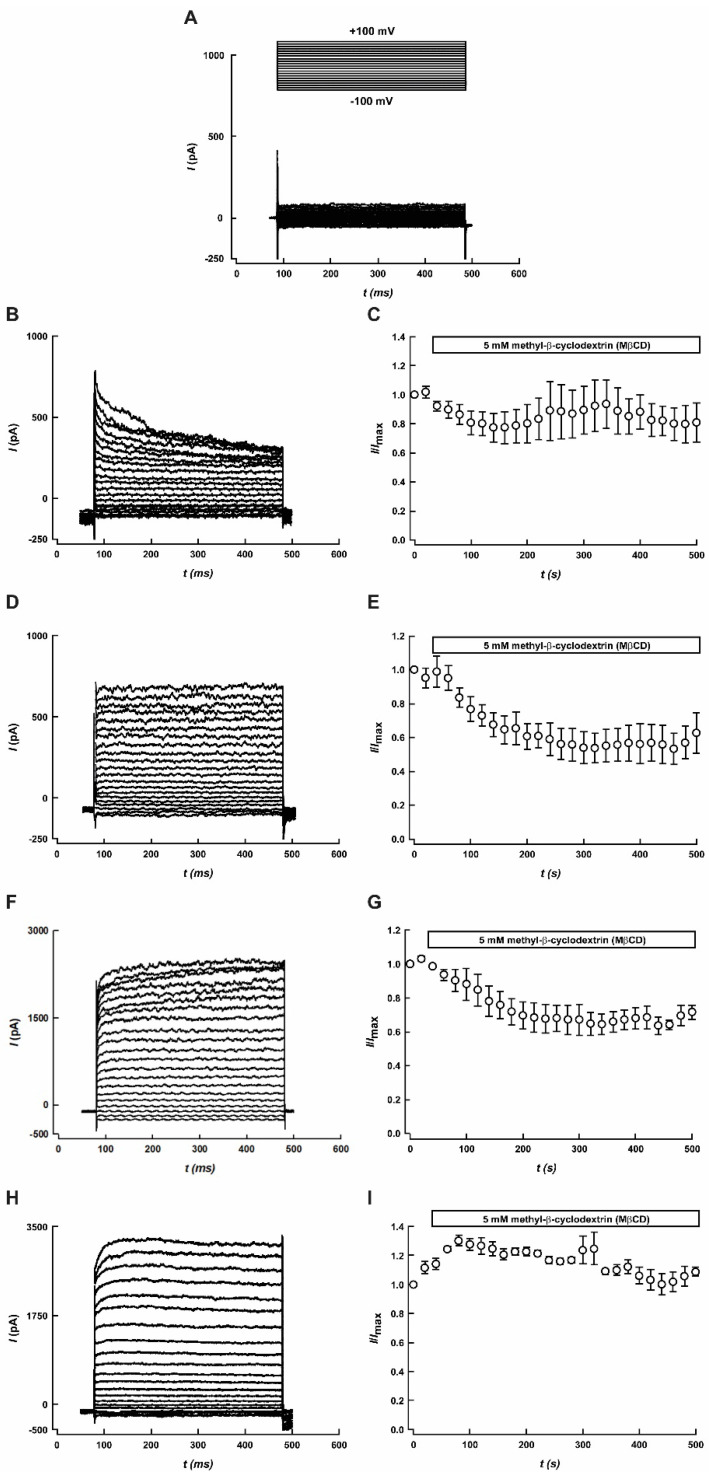
Effect of MβCD on K2P currents. (**A**) Representative recording of non-transfected HEK-293 cell. Measurements were taken using the voltage protocols (inset). (**B**,**D**,**F**,**H**) are representative current traces obtained for TWIK-1, TASK-1, TASK-3, and TRESK activities. Macroscopic K2P-mediated currents were measured using a voltage protocol consisting of 500 ms steps from −100 mV to +100 mV with increments of 10 mV and a holding potential of −80 mV. (**C**,**E**,**G**,**I**) correspond to the time course of TWIK-1, TASK-1, TASK-3, and TRESK measured at +60 mV, before and after application of 5 mM MβCD. Results are means ± SEM of at less three different experiments.

**Figure 5 biology-11-01097-f005:**
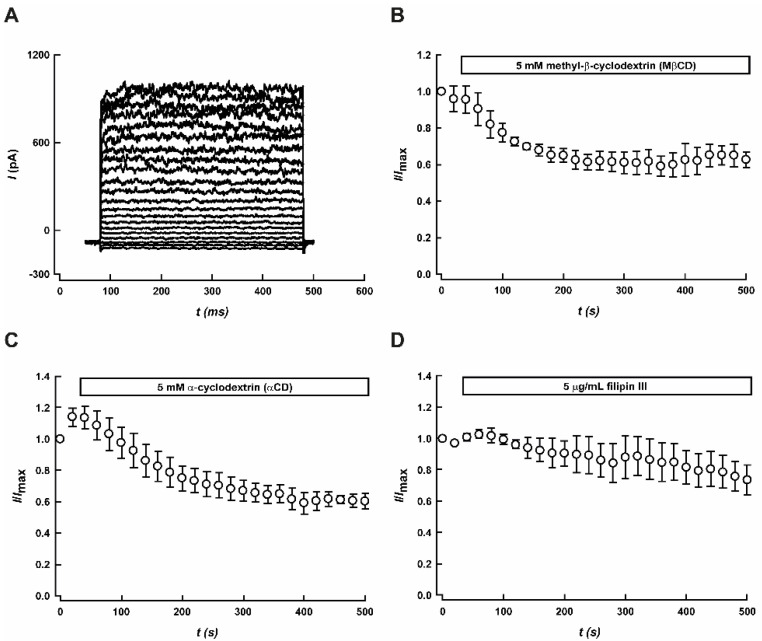
Effect of MβCD, α-cyclodextrin (αCD) and filipin III on TASK-1/TASK-3 currents expressed in HEK-293 cells. (**A**) Representative current traces obtained for TASK-1/TASK-3 concatamer, using the voltage protocol described in Figure 5. (**B**–**D**) Time course of MβCD, αCD, and filipin III treatments. Currents shown were measured at +60 mV. Results are means ± SEM of four different experiments.

**Figure 6 biology-11-01097-f006:**
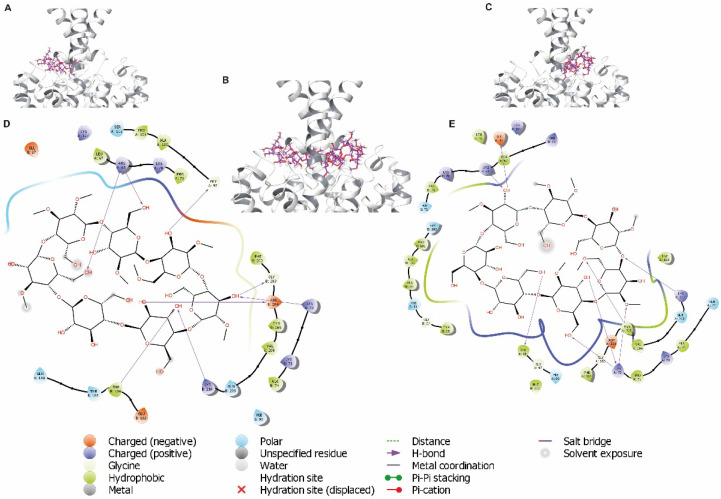
Docking predictions for MβCD in TASK-1 crystal structure. (**A**–**C**) Docking of MβCD poses (purple) located in the extracellular cavity closely positioned to Glu37, Arg68, Lys70, Gly97, Trp184, Gly203, Asp204, and Lys210 in the entry to the TASK-1 channel. (**D**,**E**) Two-dimensional interaction diagrams of MβCD in the binding sites of the TASK-1/MβCD complex in extracellular orientation. All amino acids that potentially interact with MβCD are illustrated (less than 5 Å of distance). H-bonds are represented as purple arrows. The polar and hydrophobic residues are colored in cyan and green, respectively.

**Figure 7 biology-11-01097-f007:**
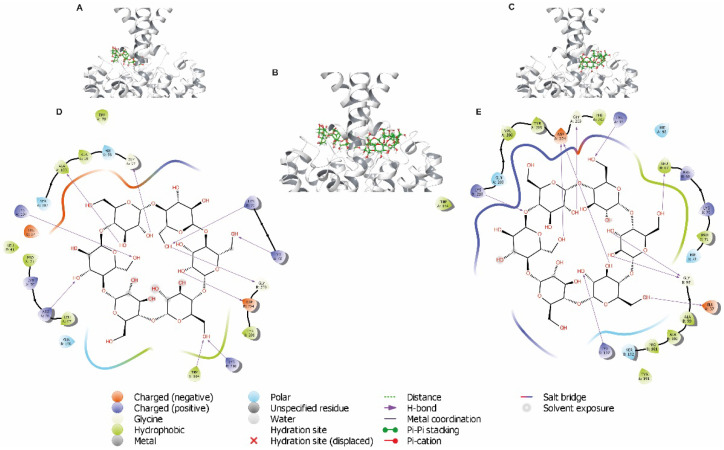
Docking analysis of αCD in the TASK-1 crystal structure. (**A**–**C**) Docking of αCD poses (green) located in the extracellular cavity closely to Glu37, Arg68, Lys70, Gly97, Trp184, Gly203, Asp204, and Lys210 in the entry to the TASK-1 channel. (**D**,**E**) Two-dimensional interaction diagrams of αCD in the binding sites of TASK-1/αCD complex in extracellular orientation. All amino acids of TASK-1 that potentially interact with αCD are illustrated. H-bonds are represented as purple arrows; cyan and green residues correspond to polar and hydrophobic, respectively.

**Figure 8 biology-11-01097-f008:**
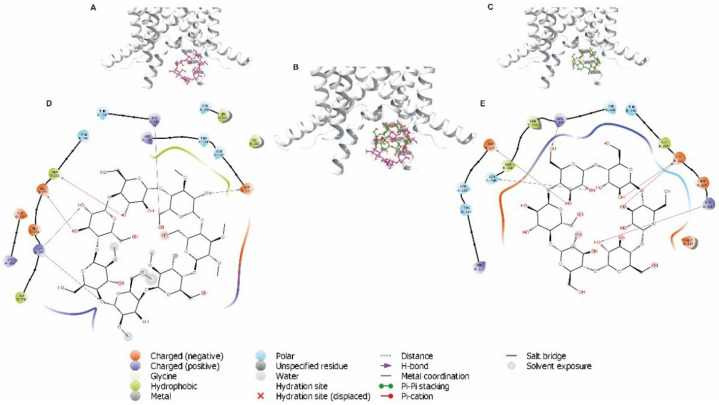
Docking predictions for MβCD and αCD in the TASK-1 crystal. (**A**–**C**) MβCD (purple) and αCD (green) located in the intracellular cavity closely to Glu130, Arg245, Glu252, and Lys255 in the TASK-1 channel, respectively. (**D**) Two-dimensional interaction diagrams of MβCD in the binding sites of TASK-1/MβCD complex in intracellular orientation. (**E**) Two-dimensional interaction diagrams of αCD in the binding sites of TASK-1/αCD complex in intracellular orientation. TASK-1 amino acids residues that potentially interact with MβCD and αCD are illustrated. H-bonds are represented as purple arrows; polar and hydrophobic residues are colored in cyan and green, respectively.

**Table 1 biology-11-01097-t001:** Input resistance and membrane potential of CGN exposed to MβCD.

	Control	MβCD
Input Resistance (MΩ)	189.63 ± 47.55	280.33 ± 48.71 *
Membrane potential (mV)	–67.49 ± 5.06	−63.47 ± 6.83

Membrane potential (mV) was determined from current ramps and it is defined as the zero-current potential. The input resistance was obtained according to the Ohm’s law from the slope of currents elicited in response to voltage ramps between +10 mV to −10 mV to the resting membrane potential. The values are represented as mean ± SEM. *: indicates statistically significant differences (*p* < 0.05) by paired *t*-test.

## Data Availability

Not applicable.

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
