# Peer review of "A Direct Interaction between Cyclodextrins and TASK Channels Decreases the Leak Current in Cerebellar Granule Neurons"

_biology, 2022, doi:10.3390/biology11081097_

Round 1

Reviewer 1 Report

The study by Zuniga et al. entitled "A direct Interaction between cyclodextrins and TASK channels degreases the leak current in cerebellar granule neurons" details a study of the effects cyclodextrin (ramdomly subsittuted M-beta-CD and alpha-CD) on the leak current of TASK channels. The science is sound and the work is well presented. Minor improvements of the english language is recommended. 

With respect to the molecular modelling simulation of the interaction between M-b-CD and the TASK-1 channel, please provide information on how the specific M-b-CD was chosen, as M-b-CD is a mixture of several hundred thousands isomers. Which isomer was chosen for the study and why?

Author Response

Response: We agree with the reviewer that there are potentially hundreds or thousands of variations of cyclodextrins that have different ring sizes and random or site-specific chemical substitutions. Furthermore all the commercially available methylated b-CDs are mixtures of various isomers and homologues except trimethyl b-CD, and the degree of substitution of commercially available methylated b-CDs falls in the range of 4–21.

In our analysis for the molecular modelling simulation, the conformation of MβCD was extracted from the crystal structure of the extracellular domain of human gastric inhibitory polypeptide receptor (PDB 2QKH) that was cocrystallized with MβCD (Parthier et al., 2007).

We have introduced the following paragraph: “The conformation of MβCD was extracted from the crystal structure of the extracellular domain of human gastric inhibitory polypeptide receptor (PDB 2QKH) that was cocrystallized with MβCD (Parthier et al., 2007)”. Lines 195–197, in the corrected version of the manuscript.

Reviewer 2 Report

In the manuscript "A Direct Interaction between Cyclodextrins and TASK Channels Decreases the Leak Current in Cerebellar Granule Neurons", Zuniga et al have show that neuronal background potassium channels are enriched in lipid rafts. While the results presented are interesting there are some concerns that need to be addressed. Overall the study merits a publication in Biophysica with minor updates.

Concerns/Comments:

1. The authors mention that the extracellular solution for neuronal patch clamp recordings has TTX to block sodium currents, what about the other channels including voltage gated calcium channels? HCN channels ?How do the authors rule out the contribution from these channels ?

2. Are the representative traces shown in fig 2 from the same cell ? If different cells current density should be plotted.

3. In figure 4 , a control of untransfected cell/cells also must be done and shown to see the contribution of endogenous channels.

4. In figure 4 and 5 It is also not immediately apparent whether the effects show after the cyclodextrin addition are indeed due to cyclodextrin or channel current rundown. Can the author show more time points with just the channel current before the addition of cyclodextrin ?

5. Can the authors demonstrate the specificity of the effects of cyclodextrin by addition of an inhibitor ? especially in the HEK cells.

Author Response

Reviewer 2

In the manuscript "A Direct Interaction between Cyclodextrins and TASK Channels Decreases the Leak Current in Cerebellar Granule Neurons", Zuniga et al have show that neuronal background potassium channels are enriched in lipid rafts. While the results presented are interesting there are some concerns that need to be addressed. Overall the study merits a publication in Biophysica with minor updates.

  1. The authors mention that the extracellular solution for neuronal patch clamp recordings has TTX to block sodium currents, what about the other channels including voltage gated calcium channels? HCN channels? How do the authors rule out the contribution from these channels?

Response: The point raised by the reviewer is a relevant one. But, In general the protocol used is widely described to record the leak potassium current, this is a non-inactivating current activated by depolarization, and that exhibited outward rectification and reversed close to the potassium equilibrium potential (–67 mV). The kinetics of activation and deactivation is very rapid (contrasting with the slow kinetics reported for the HCN channels), and all of the functional properties of IKSO correlate well with those of the K2P channels. For instance, IKSO currents are open rectifiers which are insensitive to TEA and 4-AP, blocked by Ba2+, muscarinic receptor activation, anandamide and external acidification, enhanced by halothane. The properties of this current mean that it cannot be considered as a 'M'-current (IKm), or as a delayed rectifier or a calcium-activated K+ current, and is recognizes as a 'leak' K+ current. Also the Application of 100 mM Cd2+ (a concentration which should inhibit > 98% of the Ca2+ channel currents in CG neurons; see Pearson et al. 1993; Amico et al. 1995) did not significantly change the magnitude of IKSO (C.S. Watkins and A. Mathie J. Physiol 1996).

  1. Are the representative traces shown in fig 2 from the same cell? If different cells current density should be plotted.

Response: Effectively figure 2a corresponds to the same cell recorded before and after treatment.

  1. In figure 4, a control of untransfected cell/cells also must be done and shown to see the contribution of endogenous channels.

Response: As suggested by the reviewer, we have included in the figure 4, a representative recording of non-transfected HEK-293 cell (Figure 4A).

  1. In figure 4 and 5 It is also not immediately apparent whether the effects show after the cyclodextrin addition are indeed due to cyclodextrin or channel current rundown. Can the author show more time points with just the channel current before the addition of cyclodextrin?

Response: In general, these experiments were always carried out after a period to ensure a stable recording, to a cell that served as its own control. Thus, we wait for the stabilization of the current to avoid effects such as rundown, and the current should be maintained stable for at least two consecutive sweeps, to then start the application of cyclodextrin. As example we show a representative experiment with the inclusion of initial points.

  1. Can the authors demonstrate the specificity of the effects of cyclodextrin by addition of an inhibitor? especially in the HEK cells.

Response: As can be seen in the untransfected HEK-293 cells (New figure 4A), there is a very low endogenous activity of potassium channels, which allows us to ensure that the effect shown by the cyclodextrin treatment can only be due to a direct effect on the transfected channel.
